# Work autonomy and its associated factors among nurses working in South Gondar zone public hospitals, Amhara regional state, Northcentral Ethiopia: Institution-based cross-sectional study

Yirgalem Abere[1]*, Abraham Tsedalu Amare[1], Astewle Andargie Baye[1],
Yeshiambaw Eshetie[1], Demewoz Kefale[2], Gebrehiwot Berie Mekonnen[2],
Gebrie Kassaw Yirga[1], Mengistu Ewunetu[1], Masresha Kassaw Ewinetu[1],
Bekalu Mekonen Belay[1]

**1** Department of Adult Health Nursing, College of Health Science, Debre Tabor University, Debre Tabor,
Ethiopia, **2** Department of Pediatric and Child Health, College of Health Science, Debre Tabor University,
Debre Tabor, Ethiopia

* yirgalemabere12@gmail.com

## Abstract

### Background

Work autonomy is crucial for nurses, which allows nurses to use their expertise and understanding of the profession to make informed decisions about patient care. However, research on nurses' work autonomy is limited, particularly in low-income countries like Ethiopia, which is the focus of this study.

### Objective

This study aimed to assess the level of work autonomy and its associated factors among nurses working in public hospitals in the South Gondar Zone, Ethiopia.

### Methods

A cross-sectional study was conducted in public hospitals of the South Gondar zone from January 12 to February 20, 2024. The data was collected through a self-administered pretested questionnaire. The collected data were entered into EpiData V.4.2 and then exported to SPSS V.25 for analysis. The statistical significance of the association between outcome variables and independent variables was declared at a P-value less than 5% (0.05) at 95% CI.

### Results

The overall level of good work autonomy among nurses was 53.5% (95% CI: 53.32–53.78). Factors significantly associated with good work autonomy include receiving on-the-job training (AOR = 1.6; 95% CI: 1.0–2.5), having five years of

**Data availability statement:** All relevant data are within the paper and its Supporting Information files.

**Funding:** The author(s) received no specific funding for this work.

**Competing interests:** The authors have declared that no competing interests exist.

work experience (AOR = 1.9; 95% CI: 1.1–3.2), being bothered by lack of materials (AOR = 1.8; 95% CI: 1.1–3.0), and wishing to stay in the nursing profession (AOR = 1.8; 95% CI: 1.1–3.1).

## Conclusion

While over half of nurses reported good work autonomy, critical barriers such as insufficient resources and intent to leavethe nursing profession hindered optimal autonomy. Enhancing nurse autonomy through targeted training and improved work experience is essential to advancing nursing practice and healthcare outcomes in the region.

---

## Introduction

The ability to make clinical and organizational decisions as a member of a healthcare team while adhering to the nursing discipline regulations is known as professional autonomy [1]. Being autonomous means having the ability to make judgments on your own and take action as needed without consulting other healthcare providers [2]. However, autonomy is not about working without accountability, rules, or procedures; it allows healthcare workers to perform at their best while maintaining necessary guidelines and structure [3]. For nurses, professional autonomy is essential for a positive and healthy work environment [4]. Clinical autonomy specifically refers to nurses' ability to make independent decisions regarding patient care without the direct involvement of other healthcare professionals [5,6]. Ultimately, nurse autonomy allows nurses to use their expertise and understanding of the profession to make informed decisions about patient care [7].

In nursing, autonomy is a fundamental concept, and patients benefit from high levels of autonomy [4]. Enhancing nurse autonomy improves their expertise and experience [8], positively impacting patient care, healthcare providers, and facilities. Professional autonomy leads to improvedemployee retention, reduced turnover,higher motivation,better patient safety, and enhanced care quality. It also reduces stress, lowers patient mortality, increases job satisfaction, and helps attract and retain nurses [6,9–11]. Research shows that nurses with greater professional autonomy excel at work and are more satisfied in their roles [4]. This satisfaction translates into improved patient care and enhances the healthcare system [12]. Studies have found a strong link between nurse autonomy and patient safety [13], with better care quality and reduced mortality [14]. Additionally, autonomous nurses demonstrate management and leadership abilities to deliver quality nursing care through sound clinical judgment and productive relationships with colleagues [9]. Developing ability and competence, earning the respect and trust of physicians and other colleagues, and building positive relationships with them are all crucial for acquiring independent nursing practice [15–17].

Low work autonomy is a significant barrier to delivering quality healthcare services [8]. This is an issue that affects not only low- and middle-income countries but

even industrialized ones such as the United States and Portugal [18,19]. The negative effects of limited autonomy include poor job performance, slow engagement, low job satisfaction, high turnover, conflicts between managers and health-care professionals, substandard health services, and inefficient workflows [18,20,21]. Many nurses feel their autonomy is restricted, despite the growing professionalism in nursing and an increasing focus on accountability in clinical settings [22]. This lack of autonomy is often linked to burnout, emotional exhaustion, job dissatisfaction, and negative attitudes toward patients, all of which can ultimately lead to nurses leaving the profession [13,22]. In fact, the absence of professional autonomy contributes to stress among nurses, prompting some to seek less stressful roles in places like ICUs or administrative positions or even to quit their jobs entirely [23]. Undermining job autonomy negatively affects the health of both individuals and the organization as a whole [24].

Several factors influence the autonomy of nurses in the profession, including gender, age, marital status, salary, educational background, experience, job satisfaction, hospital location, inadequate collaboration and teamwork, and the position that nurses currently hold [2,17,18,25–31]. Other relevant factors include the work unit and the specific profession within nursing [28]. Nurses today work in a variety of specialized areas, such as surgical or medical wards. Critical care and intensive care unit (ICU) nurses typically experience a higher level of autonomy compared to nurses in other settings [32].

Achieving established nursing professionalism requires first improving the autonomy of nurses [28]. Enhancing nurses' autonomy is crucial for advancing the nursing profession, as nurses play a key role in healthcare services. However, there is limited research on nurses' autonomy, especially in low-income countries like Ethiopia. The goal of this study was to assess the level of nurses' autonomy and the factors related to it in order to better understand the state of nursing autonomy.

## Methods

### Study design, area, and period

From January 12 to February 20, 2024, we conducted an institution-based cross-sectional study involving nurses employed in public hospitals in the South Gondar Zone. South Gondar is one of the thirteen zonal administrations in the northern Amhara region of Ethiopia, covering an area of 14,095.19 square kilometers. There are 424 nurses working in ten public hospitals in the zone, including Debre Tabor Comprehensive Specialized Hospital, Addis Zemen, Ebnat, Mekan-eEyesus, Andabet, Wogeda, Woreta, NefasMewucha, Dr. AmbachewMakonnen, and MigbaruKebede Primary Hospitals.

### Study population

All licensed nurses working in all South Gondar Zone public hospitals.

**Inclusion and exclusion criteria.** The study included all licensed nurses employed at public hospitals in the South Gondar Zone at the time of data collection. However, nurses on sick leave and those with fewer than six months of work experience were excluded from the study.

**Study variables and their measurement.** Nurses' work autonomy (good/poor) is the dependent variable. Five items with five Likert scales, ranging from 1 (strongly disagree) to 5(strongly agree), were used to measure it. Responses that scored equal to or below the mean value were classified as poor, while those that scored higher than the mean were classified as having good autonomy [28,31]. Independent variables include organizational factors (recognition and reward, co-worker relationship, organizational policy, being bothered by lack of material during work, participating in update training, wishing to stay in the nursing profession, and being a member of the nurse association and supervision support) as well as sociodemographic factors (age, sex, marital status, educational status, work experience, type of profession, monthly salary, living condition, and work unit).

**Sample size determination.** The total sample size was determined by using the single population proportion formula.

$$n = \frac{\left(z_{\frac{\alpha}{2}}\right)^2 * p(1-p)}{d^2}$$

After considering the level of work autonomy in western Ethiopia (46.13%) [28], the sample size was calculated based on the following assumptions: a 95% confidence level, a margin of error of 5%, and a prevalence rate (P) of 0.461. The sample size is $(1.96)^2$ x 0.46 (1-0.46)/0.0025 = 381, and a 10% non-response rate was applied; therefore, the sample size was 419. The sample size for the second objective was calculated using Epi Info 7 software, based on the assumptions of 80% power and a 95% confidence interval. Incorporating statistically significant variables from prior research [28,31], the maximum sample size was 396. However, the source population (424) was employed as the sample size for this study (the census method was used) because it was the minimum sample size required.

**Data collection tool and procedure.** A pretested and well-structured questionnaire was used to collect data. A self-administered questionnaire that was modified from other research was employed [28,31]. Every item received a score higher than 0.7 on Cronbach's Alpha, which was used to assess the tools' internal consistency. Under the constant observation of the five BSc-trained nurses who served as supervisors and trainers, ten trained BSc nurses participated in the data collection. The lead investigator was in charge of organizing the entire data-gathering process. The Hall's professionalism assessment items pertaining to belief in autonomy were used to describe the nurse's degree of autonomy. On a Likert scale, the responses are as follows: 1 means strongly disagree, 2 means disagree, 3 means neutral, 4 means agree, and 5 means strongly agree.

## Data quality control

A pretested and validated tool was used, with continuous monitoring during the data collection process to ensure data quality. Prior to the actual data collection, a pretest was conducted with 5% of the sample population at Woldia Comprehensive Specialized Hospital from November 23–27, 2023. This pretest aimed to assess the questionnaire's readability, clarity, and completion time. Based on the feedback, the tool was improved by incorporating necessary suggestions and comments.Data collectors were trained on the study's objectives, data collection methods, instruments, procedures for ensuring data accuracy, and confidentiality protocols. Once the data was verified for accuracy, it was manually coded, cleaned, edited, and entered into EpiData V.4.2. To ensure data integrity, the data was entered twice and compared with the original source. Simple frequencies and cross-tabulations were performed to check for missing values, and outliers were carefully examined.

## Data processing and analysis

After performing manual checks for consistency and completeness, the data was coded, filtered, and imported into EpiData version 4.6. It was then exported to SPSS version 25 for analysis. Descriptive analysis was conducted by computing summary statistics and proportions. Simple frequencies, summary measures, tables, and figures were used to display the data.Based on the mean value, autonomy questions were divided into two categories: poor autonomy and poor autonomy. The mean level of work autonomy among nurses was 3.17. Good autonomy is recorded as 1 for scores greater than 3.17, while weak autonomy is recorded as 0 for scores less than 3.17. Making this level of autonomy a dependent variable, bivariable and multivariable logistic regressions were used to identify factors associated with nursing autonomy.Variables from the bivariate analysis with a p-value less than 0.2 were included in the final multivariable analysis model to control for potential confounders. The entry method was used to select these variables.Before conducting regression analysis, the assumptions of the binary logistic regression model were checked. The goodness of fit was assessed using Omnibus testing and the Hosmer-Lemeshow statistics. Multicollinearity was evaluated using the variance inflation factor (VIF). The odds ratio from the multivariable binary logistic regression model was used to measure the association. A p-value of less than 0.05 was considered statistically significant for the relationship between independent and outcome variables. The minimal dataset used for this analysis is provided as Supporting Information (S1 Data).

## Ethical considerations

The College of Health Sciences Ethical Institutional Board (IRB) of Debre Tabor University granted ethics approval for the study with reference number DTU/1097/24. The College of Health Sciences Ethical Institutional Board (IRB) of Debre Tabor University granted ethics approval for the study with reference number DTU/1097/24. Written informed consent was obtained from the study participants before the study commencement. The study did not include any names or personally identifiable information. Confidentiality was preserved by eliminating direct personal identifiers from the questionnaire, employing code numbers, storing data encrypted with a password, and not misusing or revealing their information. Participants were also informed that their involvement in the study was entirely voluntary.

## Result

### Socio-demographic and professional characteristics of participants

Of the 424 participants selected for the study, 402 completed the survey, yielding a response rate of 94.8%. The majority of participants (361; 89.8%) held a bachelor's degree in nursing, and more than half (226; 56.2%) were married. Additionally, 207 respondents (51.5%) were male, and most participants had more than five years of work experience (Table 1).

**Table 1. Socio-demographic and professional characteristics of participants in South Gondar zone public hospitals, Northcentral Ethiopia, 2024 (n = 402).**

| Variables | Category | Frequency | Percent |
|---|---|---|---|
| Working unit | Medical | 79 | 19.7 |
| | Surgical | 81 | 20.1 |
| | Orthopedics | 32 | 8.0 |
| | Pediatrics | 62 | 15.4 |
| | Emergency | 27 | 6.7 |
| | ICU | 62 | 15.4 |
| | OPD | 59 | 14.6 |
| Current position | Nurse manager | 13 | 3.2 |
| | Staff nurse | 389 | 96.8 |
| Age | <30 | 205 | 51.0 |
| | 30-39 | 174 | 43.3 |
| | ≥40 | 23 | 5.7 |
| Sex | Male | 207 | 51.5 |
| | Female | 195 | 48.5 |
| Marital status | Single | 176 | 43.8 |
| | Married | 226 | 56.2 |
| Education level | Diploma | 24 | 6.0 |
| | Bachelor's degree | 361 | 89.8 |
| | Master's degree | 17 | 4.2 |
| Work shift | Day | 206 | 51.2 |
| | Night | 196 | 48.8 |
| Work experience | ≥5 | 214 | 53.2 |
| | <5 | 188 | 46.8 |

## Personal and organizational factors of respondents

In terms of organizational and personal factors, 216 people (53.7%), or the majority of health professionals, expressed satisfaction with teamwork in public hospitals. However, a sizable percentage of medical professionals expressed dissatisfaction with organizational strategy and policy, specifically 58.7%. Additionally, 50.2% of respondents said that a shortage of materials at work affected them (Table 2).

## The level of work autonomy

The overall good work autonomy in public hospitals was determined to be 53.5% (95% CI: 53.32–53.78), indicating a generally positive perception of work autonomy among the health professionals working in the study area (Table 3, Fig 1).

**Table 2. Personal and organizational factors of the respondents working in South Gondar zone public hospitals, Northcentral Ethiopia, 2024 (n = 402).**

| Variables | Category | Frequency | Percent |
|---|---|---|---|
| Recognition and reward | Satisfied | 218 | 54.2 |
| | Dissatisfied | 184 | 45.8 |
| Organizational policy and strategy | Satisfied | 166 | 41.3 |
| | Dissatisfied | 236 | 58.7 |
| Teamwork satisfaction | Satisfied | 216 | 53.7 |
| | Dissatisfied | 186 | 46.3 |
| Is there support among member | Satisfied | 211 | 52.5 |
| | Dissatisfied | 191 | 47.5 |
| Bothered from lack of material during work | Yes | 202 | 50.2 |
| | No | 200 | 49.8 |
| Participate in update training | Yes | 189 | 47.0 |
| | No | 213 | 53.0 |
| Do you wish further education | Yes | 148 | 36.8 |
| | No | 254 | 63.2 |
| Supervisor support | Yes | 212 | 52.7 |
| | No | 190 | 47.3 |
| Wish to stay in the nursing profession | Yes | 195 | 48.5 |
| | No | 207 | 51.5 |
| Member in nurse association | Yes | 223 | 55.5 |
| | No | 179 | 44.5 |

**Table 3. Belief in one's own professional autonomy among nurses in South Gondar zone public hospitals, Northcentral Ethiopia, 2024 (n = 402).**

| Autonomy item questions | Strongly agreeNo (%) | Agree No (%) | Neutral No (%) | Disagree No (%) | Strongly disagree No (%) |
|---|---|---|---|---|---|
| I make my own decisions regarding what is going to be done in my work | 49(12.2) | 70(17.4) | 114(28.4) | 98(24.4) | 71(17.7) |
| I know that my own judgment on any matter is the final judgment | 47(11.7) | 59(14.7) | 110(27.4) | 104(25.9) | 82(20.4) |
| I have much opportunity to exercise my own judgment | 51(12.7) | 58(14.4) | 111(27.6) | 74(18.4) | 108(20.9) |
| My own decisions are not subject to review by other peoples | 43(10.7) | 27(6.7) | 72(17.9) | 111(27.6) | 149(37.1) |
| I have no enforcing boss in almost every work-related my scope of work. | 47(11.7) | 58(14.4) | 130(32.3) | 79(19.7) | 88(21.9) |

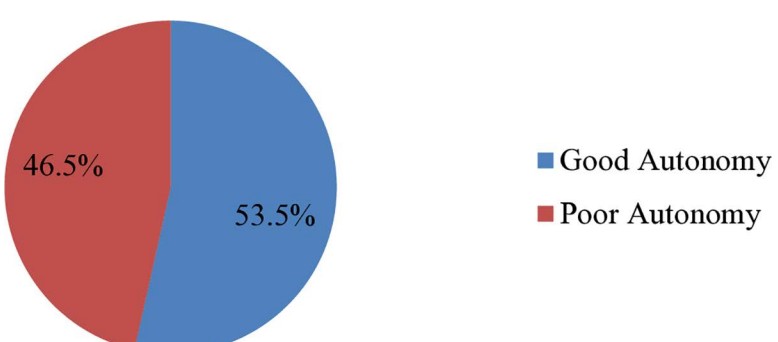

**Fig 1. Level of work autonomy among nurses in South Gondar zone public hospitals, Northcentral Ethiopia, 2024 (n=402).**

## Associated factors of work autonomy among nurses in public hospitals

In the multivariable logistic regression analysis, four variables were found to be statistically significant. Health professionals who were given job training had 1.6 times higher odds of having good work autonomy compared to those who were not given training (AOR 1.6, 95% CI 1.0–2.5). Similarly, individuals who have work experience of more than five years had 1.9 times higher odds of having good work autonomy compared to their counterparts (AOR 1.9, 95% CI 1.1–3.2). Nurses who are not botheredby a lack of material during work had 1.8 times higher odds of having good work autonomy (AOR 1.8, 95% CI 1.1–3.0). Furthermore, nurse professionals who wished to stay in the nursing profession were 1.8 times more likely to have good work autonomy compared to counterparts (AOR 1.8, 95% CI 1.1–3.1) (Table 4).

## Discussion

The findings of this study showed that 53% of nurses reported having high work autonomy; however, a significant percentage (46.5%) reported having poor work autonomy. This suggests that almost half of nurses may have trouble making decisions on their own, which affects patient treatment outcomes and care quality. Examining obstacles to autonomy and improving nurses' job autonomy are crucial. The result of this study is consistent with a study conducted on Iranian nurses that showed that greater than half of the nurses have a good level of autonomy in their work [2], Ethiopia [31],and the USA [19]. But a study conducted in Wollega Zone, Oromia, Ethiopia,showed that greater than half of the nurses have a low level of autonomy in their work [28]. Numerous factors, including the diverse study locations and periods, as well as the respondents' differing levels of education and comprehension about work autonomy in the various study contexts, could be to blame for the disparity in the degree of autonomy across that study.

Regarding the factors associated with the level of autonomy among nurses, findings from this study showed that not being bothered by a lack of materials during work is associated with having a higher level of autonomy than their counterparts. Being bothered by a lack of materials during work negatively impacts nurses' autonomy since it hinders their capacity to make prompt and wise clinical decisions [32]. Lack of resources frequently forces nurses to rely on other departments, improvise, or postpone care, all of which impair their capacity to make independent decisions. Recurring exposure to these difficulties over time might weaken nurses' self-esteem, engagement, and willingness to take independent action [33]. Furthermore, a lack of equipment hinders nurses' ability to follow standards of care and apply evidence-based practices, which lowers the quality of care provided. This finding emphasizes the necessity of providing appropriate materials to allow nurses to exercise their autonomy fully.

The results of this study also showed that study participants who received training were 1.9 times more likely to have good work autonomy as compared with their counterparts. By giving nurses the information, abilities, and self-assurance they need to make their own clinical judgments, training is essential to increasing their work autonomy. Competency is

**Table 4. Bivariable and multivariable logistic regression analysis of factors associated with work autonomy among nurses working in public hospitals of Northcentral Ethiopia, 2024 (n = 402).**

| Variables | Category | Autonomy | | COR (95%CI) | AOR(95%CI) | P-value |
|---|---|---|---|---|---|---|
| | | Good | poor | | | |
| Work experience | ≥5 | 129 | 86 | 3.3(2.2,4.9) | 1.9(1.1,3.2) | 0.018* |
| | <5 | 59 | 128 | | 1 | |
| Organizational policy and strategy | Yes | 138 | 77 | 1.6(1.1,2.4) | 1.1(0.6,1.7) | 0.846 |
| | No | 98 | 89 | 1 | 1 | |
| Team work satisfaction | Yes | 115 | 100 | 1.9(1.3,2.8) | 1.1(0.7,1.9) | 0.632 |
| | No | 71 | 116 | 1 | 1 | |
| support among member | Yes | 121 | 94 | 2.1(1.4,3.2) | 1.2(0.7,2.1) | 0.506 |
| | No | 70 | 117 | 1 | 1 | |
| on job training | Ye | 129 | 86 | 1.8(1.2,2.7) | 1.6(1.0,2.5) | .031* |
| | No | 84 | 103 | 1 | 1 | |
| Bothered from lack of material during work | No | 132 | 83 | 2.8(1.9,4.2) | 1.8(1.1,3.0) | .029* |
| | Yes | 68 | 119 | 1 | 1 | |
| Work shift | Day | 113 | 102 | 1.4(0.9,2.1) | 1.1(0.7,1.7) | 0.829 |
| | Night | 83 | 104 | 1 | 1 | |
| Recognition and reward | Yes | 108 | 107 | 1.5(1.0,2.2) | 1.4(0.8,2.4) | 0.224 |
| | No | 76 | 111 | 1 | 1 | |
| Wish further education | Yes | 148 | 67 | 1.7(1.1,2.5) | 1.2(0.7,1.9) | .510 |
| | No | 106 | 81 | 1 | 1 | |
| member in nursing association | Yes | 110 | 105 | 1.8(1.2,2.7) | 1.4(0.8,2.3) | .210 |
| | No | 69 | 118 | 1 | 1 | |
| Wish to stay in nursing profession | Yes | 140 | 75 | 3.3(2.2,5.0) | 1.8(1.1,3.1) | .026* |
| | No | 67 | 120 | 1 | 1 | |
| Supervisor support | Yes | 110 | 105 | 1.4(1.0,2.1) | 1.2(0.7,1.9) | .585 |
| | No | 80 | 107 | 1 | 1 | |

COR crude odd ratio, CI confidence interval, AOR adjusted odd ratio; 1: reference category. *Significant at p<0.05.

fostered by effective training, which empowers nurses to carry out sophisticated treatments, evaluate intricate patient needs, and apply evidence-based practices independently. Additionally, nurses are empowered to operate independently within their area of practice thanks to continual professional development, which guarantees that they remain current with healthcare trends, technology, and standards [34]. Nurses with greater training are more likely to feel secure in their skills, which enhances patient outcomes and job satisfaction.

Similarly, study participants who had work experience of more than five years were 1.8 times more likely to have good work autonomy as compared with their counterparts. This outcome is consistent with earlier studies carried out in Iran [2]. Work experience has a significant impact on nurses' autonomy at work by improving their clinical judgment, decision-making abilities, and self-assurance in handling challenging patient care scenarios. Experienced nurses get a wider understanding of interdisciplinary teamwork, healthcare systems, and patient demands in order to function more autonomously [22,35]. They gain proficiency in setting priorities, adjusting to changing clinical settings, and applying evidence-based procedures without continual supervision. Further enhancing their autonomy, seasoned nurses are frequently trusted by their peers and superiors to assume leadership positions and make important choices.

Those nurses who wish to stay in the nursing profession have a higher level of autonomy in their work than those who do not wish to stay in the nursing profession. This outcome is consistent with earlier studies carried out in Ethiopia [28].Those

that are determined to stay frequently look for positions that provide greater autonomy, decision-making authority, and chances for career advancement. Increased autonomy, improved job satisfaction, and a more proactive approach to patient care can result from this drive. On the other hand, less involved nurses could feel less autonomous since they might shy away from leadership or decision-making responsibilities.

## Limitations of this study

This study used self-administered questionnaires, which are prone to recall biases and social desirability biases. These biases may still have influenced the outcomes even if blinding was employed to lessen them. The cross-sectional technique limits the ability to infer a causal association between job autonomy and its associated attributes. The results may not be as applicable to other areas or healthcare systems because the study was limited to public hospitals in the South Gondar Zone.In this study, autonomy was measured using five Likert-type items, and scores were categorized into "good" and "poor" autonomy based on the mean value. While this approach is commonly used in similar surveys, it may oversimplify the complex and multidimensional nature of professional autonomy. This study employed a quantitative design, which limits the depth of understanding regarding the barriers to autonomy of nurses.

## Conclusion

While over half of nurses reported good work autonomy, critical barriers such as insufficient resources and intent to leave the nursing profession hindered optimal autonomy. More than five years of professional experience, on-the-job training, and a strong desire to stay in the field were all important factors linked to greater autonomy. Strengthening nurse autonomy requires addressing these issues by expanding access to resources, providing training opportunities, and encouraging sustained dedication to the field. Encouraging nurses' autonomy can result in better decision-making, increased job satisfaction, and better patient care results. We recommend future qualitative or mixed-methods studies for richer exploration of factors influencing autonomy in clinical practice, including variables such as working as nurses by choice versus not by choice.

## Supporting information

**S1 Data. Dataset: Minimal dataset used for the analysis of the study.**
(XLSX)

## Acknowledgments

We would like to express our sincere gratitude to all of the nurses who voluntarily volunteered their experiences and perspectives as part of this study. We also thank the South Gondar Zone public hospitals' administrative and medical staff for their collaboration and assistance during the study. Finally, we would like to express our sincere gratitude to our coworkers, and data collectors for their commitment and diligence during the data collection process. Your assistance was crucial in accomplishing the goals of this research.

## Author contributions

**Conceptualization:** Yirgalem Abere, Gebrehiwot Berie Mekonnen.

**Data curation:** Yirgalem Abere.

**Formal analysis:** Yirgalem Abere, Abraham Tsedalu Amare, Yeshiambaw Eshetie, Demewoz Kefale, Gebrehiwot Berie Mekonnen, Gebrie Kassaw Yirga, Masresha Kassaw Ewinetu, Bekalu Mekonen Belay.

**Funding acquisition:** Yirgalem Abere.

**Investigation:** Yirgalem Abere, Abraham Tsedalu Amare.

**Methodology:** Yirgalem Abere, Astewle Andargie Baye, Yeshiambaw Eshetie, Demewoz Kefale, Gebrehiwot Berie Mekonnen, Gebrie Kassaw Yirga, Mengistu Ewunetu, Masresha Kassaw Ewinetu, Bekalu Mekonen Belay.

**Project administration:** Yirgalem Abere.

**Resources:** Yirgalem Abere.

**Software:** Yirgalem Abere.

**Supervision:** Yirgalem Abere.

**Validation:** Yirgalem Abere.

**Visualization:** Yirgalem Abere.

**Writing – original draft:** Yirgalem Abere.

**Writing – review & editing:** Yirgalem Abere.

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
