## [Decision Letter · Decision Letter 0]

8 Aug 2025

Dear Dr. Abere,

Thank you for submitting your manuscript to PLOS ONE. After careful consideration, we feel that it has merit but does not fully meet PLOS ONE’s publication criteria as it currently stands. Therefore, we invite you to submit a revised version of the manuscript that addresses the points raised during the review process.

Please note that we have only been able to secure a single reviewer to assess your manuscript. We are issuing a decision on your manuscript at this point to prevent further delays in the evaluation of your manuscript. Please be aware that the editor who handles your revised manuscript might find it necessary to invite additional reviewers to assess this work once the revised manuscript is submitted. However, we will aim to proceed on the basis of this single review if possible.

We look forward to receiving your revised manuscript.

Kind regards,

Helen Howard

Staff Editor

PLOS ONE

Journal Requirements:

4. Please ensure that you refer to Figure 1 in your text as, if accepted, production will need this reference to link the reader to the figure.

5. We note you have included a table to which you do not refer in the text of your manuscript. Please ensure that you refer to Tables 1 and 3 in your text; if accepted, production will need this reference to link the reader to the Tables.

7. Please remove all personal information, ensure that the data shared are in accordance with participant consent, and re-upload a fully anonymized data set.

Additional guidance on preparing raw data for publication can be found in our Data Policy (https://journals.plos.org/plosone/s/data-availability#loc-human-research-participant-data-and-other-sensitive-data) and in the following article: http://www.bmj.com/content/340/bmj.c181.long .

Reviewers' comments:

Reviewer's Responses to Questions

**Comments to the Author**

1. Is the manuscript technically sound, and do the data support the conclusions?

Reviewer #1: Yes

2. Has the statistical analysis been performed appropriately and rigorously?

Reviewer #1: Yes

3. Have the authors made all data underlying the findings in their manuscript fully available?

Reviewer #1: Yes

4. Is the manuscript presented in an intelligible fashion and written in standard English?

Reviewer #1: No

Reviewer #1: Thank you for the opportunity to review this manuscript titled "Work autonomy and its associated factors among nurses working in South Gondar zone public hospitals, Amhara regional state, Northcentral Ethiopia."

Overall assessment:

The topic is relevant and helps fill a gap in the literature about professional autonomy in nursing within low-resource settings. The manuscript is generally well-organized and presents a clear research goal supported by appropriate methods. However, some areas need revision to enhance clarity and strength.

Strengths:

- Using a census-based sampling in a specific population improves the completeness of the dataset.

- The statistical approach, particularly multivariable logistic regression with proper adjustment and testing, is appropriate.

- The conclusions align with the results.

Concerns and suggestions:

1. Language and grammar:

The language used in the manuscript is clear, but several grammatical and structural issues diminish clarity (for example, “Nurse professionals who were bothered by a lack of material” – awkward phrasing). Consider professional editing to improve readability.

2. Measurement of autonomy:

The way 'autonomy' is measured with 5 Likert-type items is suitable, but dividing it into two categories (good vs. poor based on the mean) might oversimplify complex perceptions. The authors should consider discussing the limitations of using a binary outcome more thoroughly in the discussion section.

3. Limitations:

Although the limitations are discussed, I recommend emphasizing the cross-sectional design's inability to infer causality and adding a brief comment on potential self-report bias in measuring autonomy.

4. Ethics and Data Availability:

The ethics approval and consent are clearly outlined. Data availability also adheres to PLOS ONE policy.

5. Contextualization:

The introduction and discussion could benefit from better integration of relevant global literature, especially outside of sub-Saharan Africa, to expand the discussion of autonomy across healthcare systems.

Recommendation:

Minor revision — once language issues and clarification around autonomy measurement are addressed, the paper will be suitable for publication.

I appreciate the authors’ efforts and believe the paper makes a meaningful contribution to the discussion on nursing workforce development.

**Do you want your identity to be public for this peer review?** For information about this choice, including consent withdrawal, please see our Privacy Policy

Reviewer #1: No

---

## [Author Response · Author response to Decision Letter 1]

13 Aug 2025

Response to Reviewers

We appreciate the time and effort that the editors and the reviewers dedicated to providing feedback on our manuscript and are grateful for the insightful comments and valuable improvements to our paper. We have incorporated all suggestions made by the editors and the reviewers. Please see below for a point-by-point response to the reviewers’ comments and concerns. We are thankful to the reviewer for this valuable suggestion.

Point-by-point reply to the editors’ comments

RESPONSE: Thank you for your insightful recommendations. We prepared the title/authors/affiliations page and main body of the manuscript in accordance with the PLOS ONE style templates. Tables/figures are arranged and quoted sequentially.

RESPONSE: Thank you for your insightful recommendations. We expanded the ethics statement in the methods section and updated the online submission information to clearly indicate that written informed consent was obtained from the study participants before the study commencement. The study did not include any names or personally identifiable information.

RESPONSE: Thank you for your insightful recommendations. The ethics statement now appears only in the Methods section and has been removed from all other sections of the manuscript.

4. Please ensure that you refer to Figure 1 in your text as, if accepted, production will need this reference to link the reader to the figure.

RESPONSE: Thank you for your insightful recommendations. We inserted Figure 1 in the text, ensuring it appears in the correct numerical order in the manuscript.

5. We note you have included a table to which you do not refer in the text of your manuscript. Please ensure that you refer to Tables 1 and 3 in your text; if accepted, production will need this reference to link the reader to the tables.

RESPONSE: Thank you for your insightful recommendations. We inserted explicit in-text citations for Tables 1 and 3 at their first mention in the results section.

RESPONSE: Thank you for your insightful recommendations. A supporting information section has been added at the end of the manuscript, with captions for each supporting information file.

7. Please remove all personal information, ensure that the data shared are in accordance with participant consent, and re-upload a fully anonymized data set.

RESPONSE: Thank you for your insightful recommendations. We checked the dataset to confirm that all personal and indirect identifiers have been removed (not hidden).

RESPONSE: Thank you for your insightful recommendations. We looked at all of the recommended publications. Citations were included only if they were directly relevant to our work, in compliance with PLOS ONE's editorial guidelines.

Point-by-point reply to the reviewers’ comments

1. Language and grammar: The language used in the manuscript is clear, but several grammatical and structural issues diminish clarity (for example, “Nurse Professionals who were bothered by a lack of material” – awkward phrasing). Consider professional editing to improve readability. anguage and grammar:

RESPONSE: We are thankful to the reviewer for this valuable suggestion. We carefully reviewed the work for grammar and overall readability. We also carried out a thorough language edit to guarantee clarity and consistency.

2. Measurement of autonomy:

The way 'autonomy' is measured with 5 Likert-type items is suitable, but dividing it into two categories (good vs. poor based on the mean) might oversimplify complex perceptions. The authors should consider discussing the limitations of using a binary outcome more thoroughly in the discussion section.

RESPONSE: Thank you for your insightful comments and recommendations. We agree on this crucial aspect. In the revised discussion section, we have added a paragraph discussing the potential oversimplification caused by dichotomizing autonomy ratings.

3. Limitations:

Although the limitations are discussed, I recommend emphasizing the cross-sectional design's inability to infer causality and adding a brief comment on potential self-report bias in measuring autonomy.

RESPONSE: Thank you for your comments regarding the limitation section. We extended the Limitations section to emphasize that the cross-sectional design limits causal inference. We also included a note about the risk of self-report bias influencing autonomy evaluation, given that responses were based on participants' subjective views and recalls.

4. Ethics and Data Availability:

The ethics approval and consent are clearly outlined. Data availability also adheres to PLOS ONE policy.

RESPONSE: We appreciate your positive feedback.

5. Contextualization:

The introduction and discussion could benefit from better integration of relevant global literature, especially outside of sub-Saharan Africa, to expand the discussion of autonomy across healthcare systems.

RESPONSE: Thank you for pointing this out. To give a more comprehensive view of nurse autonomy in various healthcare systems, we have examined and incorporated additional material from a variety of international contexts, including research conducted in the United States.

---

## [Decision Letter · Decision Letter 1]

20 Nov 2025

Dear Dr. Abere,

Thank you for submitting your manuscript to PLOS ONE. After careful consideration, we feel that it has merit but does not fully meet PLOS ONE’s publication criteria as it currently stands. Therefore, we invite you to submit a revised version of the manuscript that addresses the points raised during the review process.

We look forward to receiving your revised manuscript.

Kind regards,

Philipos Petros Gile, MA

Academic Editor

PLOS ONE

Journal Requirements:

Reviewers' comments:

Reviewer's Responses to Questions

**Comments to the Author**

Reviewer #1: All comments have been addressed

Reviewer #2: (No Response)

2. Is the manuscript technically sound, and do the data support the conclusions?

Reviewer #1: Yes

Reviewer #2: Partly

3. Has the statistical analysis been performed appropriately and rigorously?

Reviewer #1: Yes

Reviewer #2: Yes

4. Have the authors made all data underlying the findings in their manuscript fully available?

Reviewer #1: Yes

Reviewer #2: Yes

5. Is the manuscript presented in an intelligible fashion and written in standard English?

Reviewer #1: Yes

Reviewer #2: Yes

Reviewer #1: Thank you for your thorough revision of the manuscript and your detailed point-by-point responses. The revised version demonstrates substantial improvements across all key areas.

Reviewer #2: Authors research on “Work autonomy and its associated factors among nurses working in South Gondar zone public hospitals, Amhara regional state, Northcentral Ethiopia: institution-based cross-sectional study” is interesting and valuable area of research for work productivity and mental and psychological health for both the nurses and the patients they cared for. I have a few comments authors need to address before the paper is accepted for publication.

Comments

Methods section

1.The papers would be benefited if qualitative study is included as it would allow for deeper understanding of the barriers to nurses' autonomy, making a mixed methods approach the most effective way to investigate this issue.

2. What is the relevance of stating this unless otherwise the authors believe it has something related with nurses autonomy “It is bordered to the east by the South and North Wollo zones, to the west by Lake Tana and the Bahirdar Liyu zone, to the north by Central Gondar, to the northeast by the Waghimra zone, and to the south by the East and West Gojjam zones. According to data from the South Gondar Zone Administrative Health Bureau, the population of South Gondar is 2,609,823, with 50.1% women and 49.9% men. The majority of the population lives in rural areas, with around 80-85% residing in rural districts and engaging in agriculture. Urbanization is growing, particularly around key towns like Debre Tabor, which is the administrative center of South Gondar, and other urban centers such as Woreta”. The methods need to be sharper and targeted to the goal of the study.

3.While the repeated use of 'data was' may not be a major issue, such minor grammatical errors can distract readers. It would be much better if authors review the paper for grammar throughout.

4.The authors included all nurses in the study area, noting that the study population was smaller than the intended sample size. However, they did not specify the estimated sample size before making this claim. There appears to be a significant inconsistency regarding the study population, sample size, and study design. If the authors were unable to reach their estimated sample size, it is unclear why they did not include nurses working outside hospitals, such as those in health centres within the zone. The use of a census in their methodology need to be clear such as is the census about all nurses in the zone or only those working in hospitals? It appears that a non-random sampling approach was used, focusing only on nurses working in hospitals, while many nurse professionals in the zone work in health centres and were not included. If authors are about to include nurses working in hospital, they need to answer why nurses working in health centres are excluded in the method section.

5.How authors define nurse? It would be clearer who is considered nurse in this study and who is included.

Registered diploma nurse, BSc nurse, mental health nurse, nurses with advanced training like Perioperative Anaesthesia Nurse and so on. This is very important particularly their work autonomy might be affected by their level of training and scope that authors may consider adjusting in their regression analysis. Perhaps authors need clearly state this in the method section.

6.Do the authors assess whether the type of job nurses aimed for affects their work autonomy? What percentage of the study participants are working as nurses by choice versus not by choice? This is a sensitive variable that may be better explored through qualitative research and should be examined further for discussion and recommendations.

7.What does the mean score refers? Was it the mean score of each item for all participants or scores from the 5 items are summed or averaged first to get an overall autonomy score for each nurse. Then calculate the mean averaged score of the participant. This needs to be clearer in the method section.

Results section

8.This needs to be entirely presented in the methods section “Using logistic regression analysis with odds ratios (ORs) and 95% confidence intervals (CIs), factors related to work autonomy at public hospitals in the South Gonder Zone, Northcentral Ethiopia, were identified. First, the relationship between each independent variable and the result variable was evaluated using bivariable logistic regression. Twelve variables with a p-value below 0.25 were selected as candidates for multivariable logistic regression analysis. Variables with a p-value less than 0.05 in the multivariable logistic regression analysis were considered statistically significant.”

Discussion

9.There is contradicting results discussed for example authors said “Nurses who reported being bothered by a lack of material during work had 1.8 times higher odds of having good work autonomy (AOR 1.8, 95% CI 1.1–3.0)” while down in the discussion they stated that “findings from this study showed that being bothered by a lack of materials during work is associated with having a lower level of autonomy than their counterparts”. Authors needed to be clearer and careful of interpreting the findings.

10. The result showed 53.5 % of nurses had good work autonomy (above the mean score). However, the authors should also consider the reverse perspective, as 46.5% of nurses report poor work autonomy, a substantial proportion that warrants further investigation and needs to be discussed.

I would be happy to review this paper after authors revise it according to the comments.

**Do you want your identity to be public for this peer review?** For information about this choice, including consent withdrawal, please see our Privacy Policy

Reviewer #1: No

Reviewer #2: No

---

## [Author Response · Author response to Decision Letter 2]

24 Nov 2025

Response to Reviewers

We appreciate the time and effort that the editors and the reviewers dedicated to providingfeedback on our manuscript and are grateful for the insightful comments and valuableimprovements to our paper. We have incorporated all suggestions made by the reviewers. Please see below for a point-by-pointresponse to the reviewers’ comments and concerns. We are thankful to the reviewer for this valuable suggestion.

Point-by-point reply to the reviewers’ comments

Reviewer #1

Abstract

1.The papers would be benefited if qualitative study is included as it would allow for deeper understanding of the barriers to nurses' autonomy, making a mixed methods approach the most effective way to investigate this issue.

RESPONSE: Thank you for your comments and recommendations. We agreed that incorporating qualitative methods allows for deeper understanding of the barriers to nurses' autonomy. While the current study was quantitative, we acknowledge this limitation in the conclusion and limitation section and recommend future studies to consider a mixed-method approach.

2. What is the relevance of stating this unless otherwise the authors believe it has something related with nurses autonomy “It is bordered to the east by the South and North Wollo zones, to the west by Lake Tana and the Bahirdar Liyu zone, to the north by Central Gondar, to the northeast by the Waghimra zone, and to the south by the East and West Gojjam zones. According to data from the South Gondar Zone Administrative Health Bureau, the population of South Gondar is 2,609,823, with 50.1% women and 49.9% men. The majority of the population lives in rural areas, with around 80-85% residing in rural districts and engaging in agriculture. Urbanization is growing, particularly around key towns like Debre Tabor, which is the administrative center of South Gondar, and other urban centers such as Woreta”. The methods need to be sharper and targeted to the goal of the study.

RESPONSE: Thank you for your comments and recommendations on improving our manuscript. We omitted the thorough geographic and demographic description of the South Gondar Zone from the revised manuscript since it was not relevant to the study's purpose.

3. While the repeated use of 'data was' may not be a major issue, such minor grammatical errors can distract readers. It would be much better if authors review the paper for grammar throughout.

RESPONSE: We are thankful to the reviewer for this valuable suggestion. We appreciate grammatical concerns and clarity; we have revised the entire manuscript for grammatical accuracy.

4. The authors included all nurses in the study area, noting that the study population was smaller than the intended sample size. However, they did not specify the estimated sample size before making this claim. There appears to be a significant inconsistency regarding the study population, sample size, and study design. If the authors were unable to reach their estimated sample size, it is unclear why they did not include nurses working outside hospitals, such as those in health centres within the zone. The use of a census in their methodology need to be clear such as is the census about all nurses in the zone or only those working in hospitals? It appears that a non-random sampling approach was used, focusing only on nurses working in hospitals, while many nurse professionals in the zone work in health centres and were not included. If authors are about to include nurses working in hospital, they need to answer why nurses working in health centres are excluded in the method section.

RESPONSE: We are thankful to the reviewer for this valuable suggestion. We have specified the estimated sample size and all suggestions given in the sample size determination section of the revised manuscript based on your comments.

5. How authors define nurse? It would be clearer who is considered nurse in this study and who is included. Registered diploma nurse, BSc nurse, mental health nurse, nurses with advanced training like Perioperative Anaesthesia Nurse and so on. This is very important particularly their work autonomy might be affected by their level of training and scope that authors may consider adjusting in their regression analysis. Perhaps authors need clearly state this in the method section.

RESPONSE: We are thankful to the reviewer for this valuable comment. In this study "nurses" refers to all licensed nurse professionals.

6. Do the authors assess whet her the type of job nurses aimed for affects their work autonomy? What percentage of the study participants are working as nurses by choice versus not by choice? This is a sensitive variable that may be better explored through qualitative research and should be examined further for discussion and recommendations.

RESPONSE: Thank you for your insightful comments and recommendations. We did not specifically assess whether the type of job nurses aimed for (working as nurses by choice versus not by choice) influenced their work autonomy. We have now acknowledged this limitation in the limitation section of the revised manuscript.

7. What does the mean score refers? Was it the mean score of each item for all participants or scores from the 5 items are summed or averaged first to get an overall autonomy score for each nurse. Then calculate the mean averaged score of the participant. This needs to be clearer in the method section.

RESPONSE: RESPONSE: Thank you for your comments. We acknowledge our description of how the autonomy score was calculated was not sufficiently clear. We have now clarified the scoring procedure in the methods section.

Results

8. This needs to be entirely presented in the methods section “Using logistic regression analysis with odds ratios (ORs) and 95% confidence intervals (CIs), factors related to work autonomy at public hospitals in the South Gonder Zone, Northcentral Ethiopia, were identified. First, the relationship between each independent variable and the result variable was evaluated using bivariable logistic regression. Twelve variables with a p-value below 0.25 were selected as candidates for multivariable logistic regression analysis. Variables with a p-value less than 0.05 in the multivariable logistic regression analysis were considered statistically significant.”

RESPONSE: Thank you for your comments. We are now incorporate this in the methods section and remove in the result section to avoid reputations.

Discussion

9. There is contradicting results discussed for example authors said “Nurses who reported being bothered by a lack of material during work had 1.8 times higher odds of having good work autonomy (AOR 1.8, 95% CI 1.1–3.0)” while down in the discussion they stated that “findings from this study showed that being bothered by a lack of materials during work is associated with having a lower level of autonomy than their counterparts”. Authors needed to be clearer and careful of interpreting the findings.

RESPONSE: Thank you for insightful comments. We have corrected this inconsistency to ensure accurate interpretation of the findings

10. The result showed 53.5 % of nurses had good work autonomy (above the mean score). However, the authors should also consider the reverse perspective, as 46.5% of nurses report poor work autonomy, a substantial proportion that warrants further investigation and needs to be discussed.

RESPONSE: Thank you for pointing this out. We have revised the discussion section to highlight 46.5% of nurses report poor work autonomy.

---

## [Decision Letter · Decision Letter 2]

3 Dec 2025

Work autonomy and its associated factors among nurses working in South Gondar zone public hospitals, Amhara regional state, Northcentral Ethiopia: institution-based cross-sectional study

PONE-D-25-21650R2

Dear Author,

We’re pleased to inform you that your manuscript has been judged scientifically suitable for publication and will be formally accepted for publication once it meets all outstanding technical requirements.

Kind regards,

Philipos Petros Gile, MA

Academic Editor

PLOS ONE

Additional Editor Comments (optional):

Reviewers' comments:

Reviewer's Responses to Questions

**Comments to the Author**

Reviewer #2: All comments have been addressed

2. Is the manuscript technically sound, and do the data support the conclusions?

Reviewer #2: Partly

3. Has the statistical analysis been performed appropriately and rigorously?

Reviewer #2: Yes

4. Have the authors made all data underlying the findings in their manuscript fully available?

Reviewer #2: Yes

5. Is the manuscript presented in an intelligible fashion and written in standard English?

Reviewer #2: No

Reviewer #2: The methods section needs further work to make the manuscript tight enough for the scientific audience.

**Do you want your identity to be public for this peer review?** For information about this choice, including consent withdrawal, please see our Privacy Policy

Reviewer #2: No

---

## [Editor Report · Acceptance letter]

PONE-D-25-21650R2

PLOS One

Dear Dr. Abere,

I'm pleased to inform you that your manuscript has been deemed suitable for publication in PLOS One. Congratulations! Your manuscript is now being handed over to our production team.

Kind regards,

on behalf of

Dr. Philipos Petros Gile

Academic Editor

PLOS One